# Importation of Alpha and Delta variants during the SARS-CoV-2 epidemic in Switzerland: Phylogenetic analysis and intervention scenarios

Martina L. Reichmuth[1]☺*, Emma B. Hodcroft[1,2,3]☺, Christian L. Althaus[1,3]

1 Institute of Social and Preventive Medicine, University of Bern, Bern, Switzerland, 2 Swiss Institute of Bioinformatics, Lausanne, Switzerland, 3 Multidisciplinary Center for Infectious Diseases, University of Bern, Bern, Switzerland

☺ These authors contributed equally to this work.
* martina.reichmuth@unibe.ch

**Data Availability Statement:** The data of confirmed SARS-CoV-2 cases are openly shared by the Swiss Federal Office of Public Health (FOPH) and sequence data are available on GISAID after

## Abstract

The SARS-CoV-2 pandemic has led to the emergence of various variants of concern (VoCs) that are associated with increased transmissibility, immune evasion, or differences in disease severity. The emergence of VoCs fueled interest in understanding the potential impact of travel restrictions and surveillance strategies to prevent or delay the early spread of VoCs. We performed phylogenetic analyses and mathematical modeling to study the importation and spread of the VoCs Alpha and Delta in Switzerland in 2020 and 2021. Using a phylogenetic approach, we estimated between 383–1,038 imports of Alpha and 455–1,347 imports of Delta into Switzerland. We then used the results from the phylogenetic analysis to parameterize a dynamic transmission model that accurately described the subsequent spread of Alpha and Delta. We modeled different counterfactual intervention scenarios to quantify the potential impact of border closures and surveillance of travelers on the spread of Alpha and Delta. We found that implementing border closures after the announcement of VoCs would have been of limited impact to mitigate the spread of VoCs. In contrast, increased surveillance of travelers could prove to be an effective measure for delaying the spread of VoCs in situations where their severity remains unclear. Our study shows how phylogenetic analysis in combination with dynamic transmission models can be used to estimate the number of imported SARS-CoV-2 variants and the potential impact of different intervention scenarios to inform the public health response during the pandemic.

## Author summary

We were interested in quantifying the number of imports of SARS-CoV-2 variants of concern (VoCs) and assessing the potential impact of travel restrictions and surveillance strategies in Switzerland. We used genomic surveillance data to calculate when and how often two different VoCs, Alpha and Delta, were imported into Switzerland. We used these

registration, as EPI_SET_221003xn (S1 Table). Most Swiss sequences (93%) that we used are also available openly, see S2 Table for list of accession numbers. Our code is openly accessible in the following repositories: github.com/ISPMBern/voc_imports_ch (https://doi.org/10.5281/zenodo.7994708) for the transmission modeling, https://github.com/emmahodcroft/ncov_2021/tree/random_context_reduce (https://doi.org/10.5281/zenodo.7970773) for the phylogenetic analysis and github.com/emmahodcroft/Intros-CH-AlphaDelta (https://doi.org/10.5281/zenodo.7970756) for the inference of importations.

**Funding:** MLR and CLA were supported by the European Union's Horizon 2020 research and innovation program - project EpiPose (No 101003688). CLA and EBH received funding from the Swiss National Science Foundation (No 196046). The funder had no role in the study design, data collection and analysis, decision to publish, or preparation of the manuscript.

**Competing interests:** The authors have declared that no competing interests exist.

estimates to simulate the spread of VoCs in a transmission model and investigated counterfactual intervention scenarios. Even though there were hundreds to a thousand imports, implementing border closures following the announcement of VoCs would have had limited impact on delaying their spread. However, improved surveillance of travelers would be a more effective measure to delay the spread of VoCs. In conclusion, our study illustrates that phylogenetic analysis combined with mathematical transmission models can be used to inform the public health response during pandemics.

## Introduction

Since 2019, the severe acute respiratory syndrome coronavirus type 2 (SARS-CoV-2) has been spreading continuously, which has driven its evolution. Variants of concern (VoCs) have emerged which are associated with increased transmissibility, immune escape, changes in disease severity, or a combination thereof [1]. Alpha and Delta emerged in late 2020 and mid 2021, and their transmission dynamics were influenced by importations, the heterogeneous landscape of naturally acquired and vaccine-elicited immunity and non-pharmaceutical interventions (NPIs) such as surveillance of travelers and travel restrictions. It was previously shown that the resumption of travel in summer 2020 after strict control interventions which—while varying by country—often involved border closures, working from home and restricting social contacts, led to the importation of a new SARS-CoV-2 variant that fueled epidemics in countries with low-incidence [2,3]. In winter 2020/2021, Swiss authorities tightened travel restrictions again for travelers from the United Kingdom (UK) [4], which were followed by overall stricter measures in Switzerland once concerns about the newly detected Alpha variant were communicated. To date, however, the impact of such travel restrictions and surveillance strategies to prevent or delay the introduction of VoCs is not well understood.

Global genomic surveillance is vital to capture the transmission dynamics of different SARS-CoV-2 variants around the world. Genomic and epidemiological data allow characterization and quantification of the spread of VoCs [5–9]. At the end of 2020, deletions in the Alpha variant meant that Polymerase Chain Reaction (PCR) testing methods targeting a particular section of the S gene failed, leading to 'S gene target failure' (SGTF), which allowed tracking of the spread of Alpha, and in some cases, preferential sequencing of Alpha cases [7]. Alpha, which was initially detected in the UK, showed a 37%-100% increase in transmissibility over previously circulating variants [5,7–11]. In early 2021, Alpha was continuously displaced by another VoC, Delta, which likely emerged in India [12]. For Delta, studies have estimated an increased transmissibility compared to previously circulating variants, which were predominantly Alpha in Europe and North America, Beta in South Africa, and Gamma in South America, between 40% and 167% [8,9]. VoCs emerge in one place and spread to different countries through travel. Therefore, quantifying the impact of travel and imported SARS-CoV-2 cases can help guide pandemic responses, particularly prior to wide-spread vaccination. Routinely obtained surveillance data can provide valuable information about the place of exposure, but can be biased [13]. Leveraging the wealth of genomic data offers the opportunity to integrate an additional level of information to inform the impact of travel and importation events. To try to prevent or slow the import of VoCs, some countries have introduced travel restrictions such as vaccination certificates, testing, or travel bans. Although travel restrictions have not prevented the global spread of VoCs, they could potentially mitigate the early spread and lessen the impact of new VoCs on hospitalizations and deaths until there is an improved understanding of new VoCs to then intervene appropriately.

Mathematical modeling studies and phylogenetic analyses are central to understand the transmission dynamics and evolution of infectious diseases and to inform public health decision making [14]. For example, the early description of the emergence of Alpha was based on mathematical and statistical modeling approaches and critically influenced the early response against the variant [10]. Phylodynamic analyses have been used to study the evolution of infectious pathogens in intra- and inter-host systems [15,16]. For example, BEAST [17] can be combined with compartmental transmission models to infer epidemiological parameters such as the basic reproduction number $R_0$ [18,19]. Inter-host epidemiological dynamics have also been used to study the importation of infectious pathogens such as Zika virus, dengue virus or SARS-CoV-2 [3,6,20–26]. Integrating phylogenetic analyses with dynamic transmission models also has the potential to provide comprehensive inferences and projections of the SARS-CoV-2 pandemic.

In this study, we used a stepwise approach of phylogenetic analyses in combination with a dynamic SARS-CoV-2 transmission model to study the spread of VoCs in Switzerland. First, we reconstructed phylogenies of the Alpha and Delta variants to estimate the number of imported cases into Switzerland. Second, we modeled the importation and spread of Alpha and Delta in Switzerland. Finally, we simulated the impact of three counterfactual scenarios to mitigate the spread of Alpha and Delta.

## Methods

### Data

We used publicly available data from the Swiss Federal Office of Public Health (FOPH), CoV-Spectrum, and the Federal Statistical Office (FSO) on daily laboratory-confirmed SARS-CoV-2 cases, daily estimates of the effective reproduction number ($R_e$), genomic metadata, and the population size [27–32]. We accessed sequencing data via GISAID [33] on 15 February 2022, including data generated in Switzerland as part of a federal consortium [34], to ensure the considered sequences encompass the introduction periods of Alpha and Delta into Switzerland. We used a combination of code from CoVariants.org and in-house scripts (available at https://github.com/emmahodcroft/Intros-CH-AlphaDelta) to select all Alpha (Nextstrain clade 20I) and Delta (Nextstrain clades 21A, 21I, and 21J) sequences sampled in Switzerland prior to 31 March 2021 and 31 July 2021, respectively, which are the dates when the proportion of each VoC was over 90% among all sequenced cases in Switzerland. We then ran the selected sequences through the Nextstrain ncov pipeline [35], as modified for CoVariants.org, to generate 'focal' phylogenies (available at https://github.com/emmahodcroft/ncov_2021/tree/random_context_reduce). We minimized bias and improved phylogenetic inference by masking highly homoplastic sites and removing sequences with fewer than 27,000 bases, as recommended by de Maio et al. (2020) [36]. These focal trees contained all Alpha or Delta sequences from Switzerland before the 90% VoC representation date which passed quality control criteria. Alongside these, to maximize the chance of including closely-related non-Swiss sequences, which allows detection of a putative importation, we included up to 10,000 non-Swiss, global 'context' sequences that were most genetically similar to our focal set. We chose the number of genetically similar sequences in order to have a dataset of approximately 15,000 to 18,000 sequences for computational tractability, while resulting in 1.25 to 2 genetically similar global sequences per focal sequence. In addition, approximately 200 random 'background' sequences, distributed through time, were included to ensure the tree rooted correctly. Both of these context and background sequence sets were generated using the algorithms in the Nextstrain ncov pipeline (see Method in S1 File), and were also sampled prior to 31 March 2021 and 31 July 2021 for Alpha and Delta respectively. The final set of sequences that we used for the analysis

can be seen on GISAID under EPI_SET_221003xn (S1 Table); 93% of the Swiss sequences are also available openly (S2 Table). As a sensitivity analysis, we randomly down-sampled all available non-Swiss sequences prior to our cutoff dates by 50% before selecting the most 10,000 genetically similar sequences, and repeated this ten times. The objective was to assess the influence of context sequencing coverage on our phylogenetic analysis and estimates of the imports.

### Phylogenetic analysis

To estimate the number of imports of VoCs into Switzerland, we collapsed phylogenetic trees into clusters. As previously described in Hodcroft et al. [3], we collapsed subtrees that contain only sequences from a single country into the parental node to form a polytomy. This process was repeated in a recursive 'bottom-up' fashion, such that every node eligible for collapsing was collapsed. After collapsing, we labeled internal nodes by the proportion of the geographic origin of their direct children (see Method in S1 File for a more detailed explanation). Because both Alpha and Delta originated outside of Switzerland, the roots of these variants are non-Swiss, and thus, we inferred a putative import whenever a node without Swiss sequences led to a node with Swiss sequences. We used two approaches to estimate the number of imports, referred to as the liberal and conservative phylogenetic approach, which infer upper and lower bounds on the number of imports. The liberal approach considered any mixed Swiss and non-Swiss node as an introduction (Fig 1A). On the other hand, when Swiss and non-Swiss sequences formed a subtree (with mixed-country nodes leading to more mixed-country

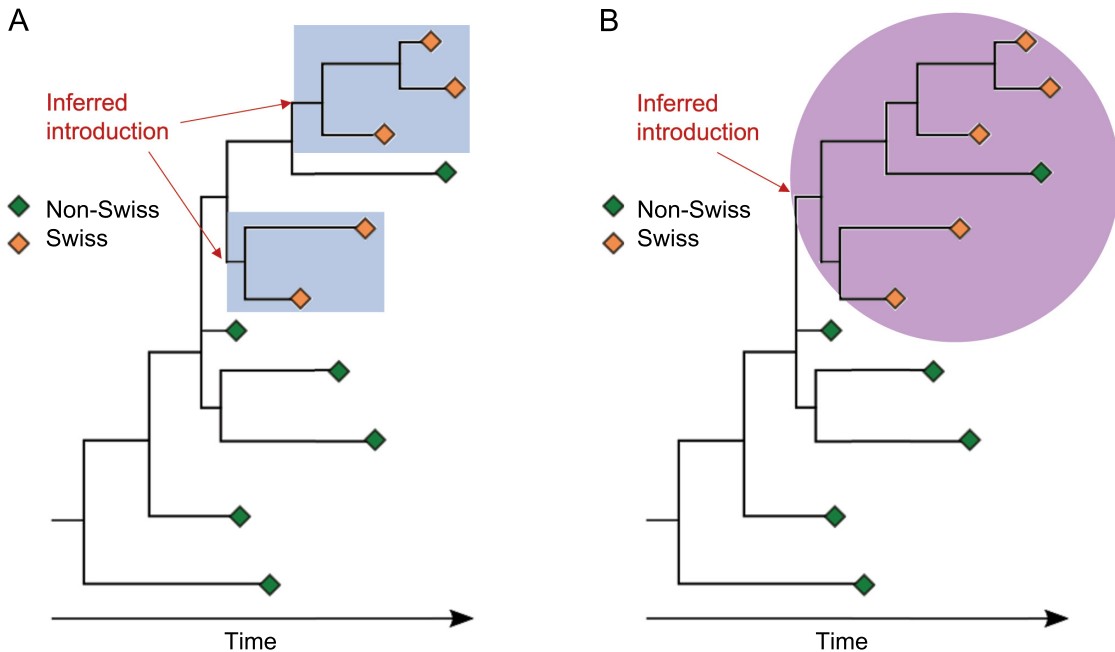

**Fig 1. Illustrative example of the conservative and liberal approach to estimate the importation of SARS-CoV-2 variants of concern (VoC) to Switzerland.** A: In the liberal approach, each subsequent subtree of only Swiss sequences is considered as a separate importation event. In this example, there would be two imports to Switzerland. B: In the conservative approach, subtrees with mixed Swiss and non-Swiss sequences are considered as one importation. Non-Swiss sequences are assumed to be exports from Switzerland or to originate from parallel evolution outside of Switzerland. In this example, there would be one import into Switzerland. More details are provided in the Method in S1 File.

nodes), the conservative approach counted all directly linked mixed-country nodes as only one import, with further non-Swiss sequences assumed to originate from parallel evolution outside of Switzerland or exports from Switzerland (Fig 1B). Import events are recorded and were then used to parameterize the transmission model (see next section about the transmission model).

The liberal approach may overestimate imports, as further exports from Switzerland to other countries or diversification within Switzerland could be wrongly considered imports. In contrast, the conservative approach considered only the earliest clusters with a Swiss sequence as imports, which means that further potential imports were strictly excluded. We chose the date of the earliest sequence of a cluster as the import date and shifted the date backwards by seven days to account for the delay from infection to case reporting, i.e., lag of detection. In a sensitivity analysis, we used a range of three to thirteen days for the lag of detection derived from du Plessis et al. (2021) [6] (Fig D in S1 File). All code used to identify the selected sequences, to analyze the resulting phylogenies, to estimate the number of imports, and for the phylogenetic builds can be found on GitHub: github.com/emmahodcroft/Intros-CH-AlphaDelta.

## Transmission model

**Deterministic model.** We used a deterministic, population-based transmission model for SARS-CoV-2 in Switzerland that was described by the following set of ordinary differential equations (ODEs) (Fig 2):

$$dS/dt = -\beta_t SI_1 - (1 + \kappa)\beta_t SI_1 - \omega_t,$$

$$dE_1/dt = \beta_t SI_1 - \sigma E_1,$$

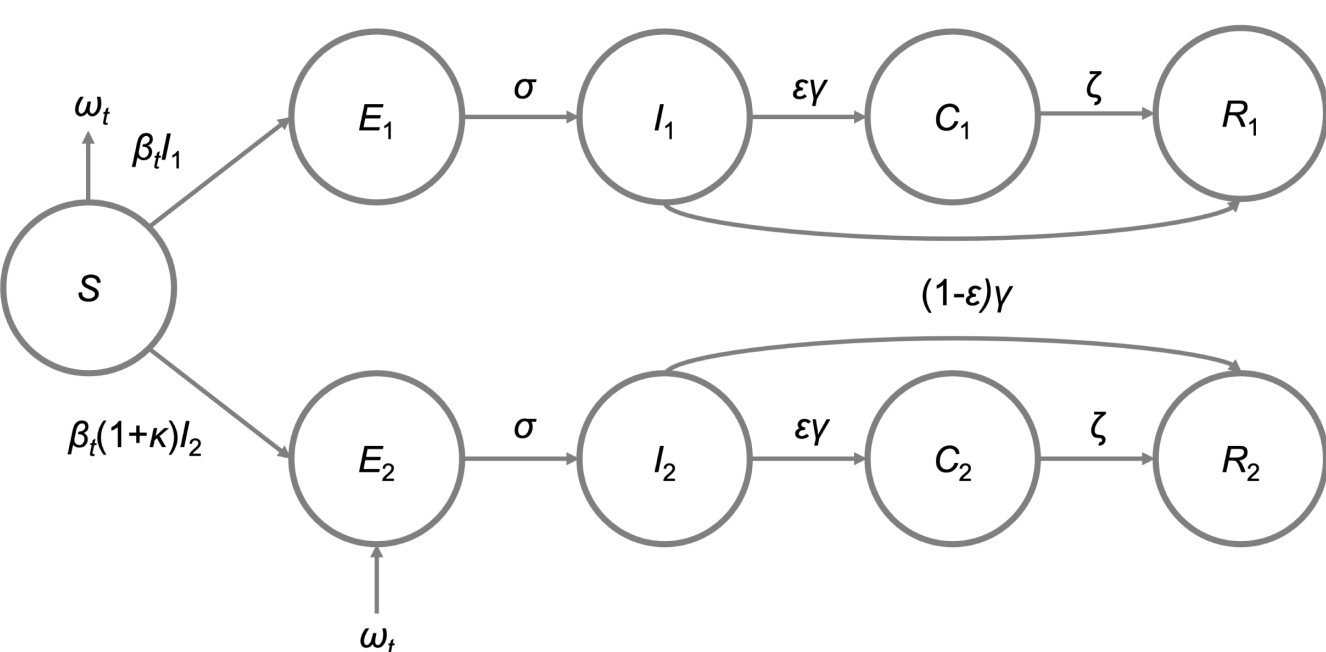

**Fig 2. Scheme of the SARS-CoV-2 transmission model.** The model includes individuals that are susceptible ($S$), exposed to the pre-circulating variant ($E_1$) or the imported VoC ($E_2$), infected with the pre-circulating variant ($I_1$) or the imported VoC ($I_2$), tested ($C_1$, $C_2$), and recovered from infection ($R_1$, $R_2$). $\omega_t$ denotes the importation of VoC as derived from the phylogenetic analysis.

$$dE_2/dt = (1 + \kappa)\beta_t SI_2 - \sigma E_2 + \omega_t,$$

$$dI_1/dt = \sigma E_1 - \gamma I_1,$$

$$dI_2/dt = \sigma E_2 - \gamma I_2,$$

$$dC_1/dt = \varepsilon \gamma I_1 - \zeta C_1,$$

$$dC_2/dt = \varepsilon \gamma I_2 - \zeta C_2,$$

$$dR_1/dt = (1 - \varepsilon)\gamma I_1 + \zeta C_1,$$

$$dR_2/dt = (1 - \varepsilon)\gamma I_2 + \zeta C_2,$$

where susceptible individuals $S$ can get infected by individuals that either carry the pre-circulating variant ($I_1$) or the imported VoC ($I_2$) at rates $\beta_t$ and $(1+\kappa)\beta_t$, respectively. $\kappa$ denotes the increased transmissibility of the imported VoC. We calculated $\kappa$ from the estimated growth advantage $\rho$ of the new VoC compared to the previously circulating variants using a logistic growth model (binomial regression) for the proportion of the new variant among all previously circulating variants (Fig E in S1 File). Assuming no change in the generation time $D$ and no immune evasion, $\kappa = \rho D/R_w$ [11], where $R_w$ is the effective reproduction number of the previously circulating variants during the time period of replacement. We sampled from that the publicly available estimates of the daily overall effective reproduction number $R_e$ from 1 November 2020 to 31 January 2021 (early growth phase of Alpha) and from 1 April 2021 to 30 June 2021 (early growth phase of Delta), assuming the values correspond to $R_w$ during these time periods (https://github.com/covid-19-Re) (Fig 3B) [28]. Additionally, we sampled from the estimated $\rho$ and calculated the median $\kappa$. We then expressed the time-dependent transmission rate as a function of the overall $R_e$ as follows:

$$\beta_t = \frac{R_e S}{(1 + p\kappa)\gamma}$$

where $p = E_2/(E_1+E_2)$ corresponds to the proportion of the imported VoC. Exposed individuals $E_1$ and $E_2$ move through an incubation period at rate $\sigma$ before they become infectious individuals $I_1$ and $I_2$ for $1/\gamma$ days. A fraction $\varepsilon$ of infected infectious individuals enter a testing compartment where they get tested at rate $\zeta$ before entering the recovered compartment, whereas the remainder $(1-\varepsilon)$ does not get tested and moves directly from the infected compartment to the recovered compartment. $\omega_t$ corresponds to the time-dependent rate of importation of VoCs. We parameterized the vector $\omega_t$ using the daily number of estimated imports from the phylogenetic analysis (Fig 4A and 4B), e.g., $\omega_t$ for Alpha on 1 February 2021 was 8 and 2 for the liberal and conservative approach, respectively.

We simulated importation of Alpha and Delta to Switzerland during the period from 1 October 2020 to 1 May 2021 and 1 February 2021 to 1 September 2021. We designated variants as either 'Alpha', 'Delta' or 'other variants' and the mathematical modeling analyses started on 1 October 2020 for Alpha and 1 February 2021 for Delta and ended on 1 May and 1 September, respectively. The initial state variables and the model parameters were informed by the literature, demography, estimates, and assumptions (Table 1). The initial number of recovered individuals infected with all previously circulating variants was informed by the Corona

**A**

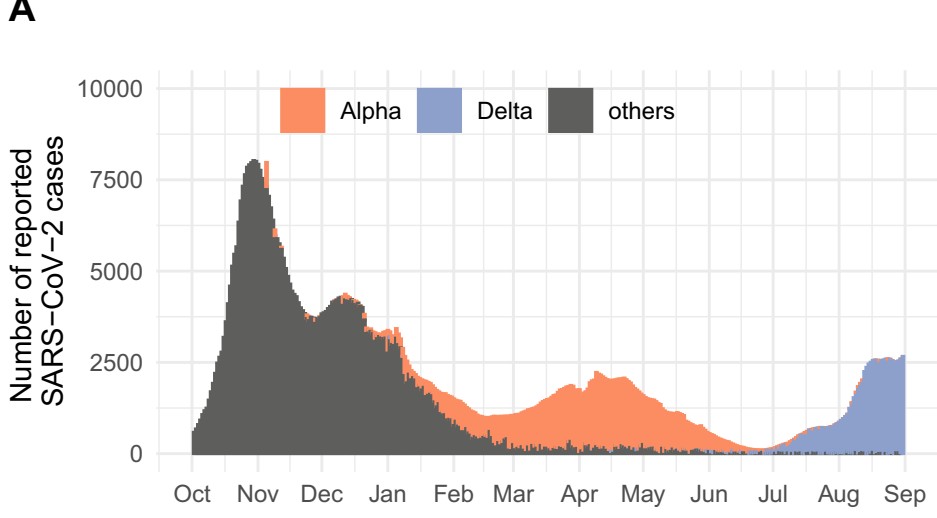

**B**

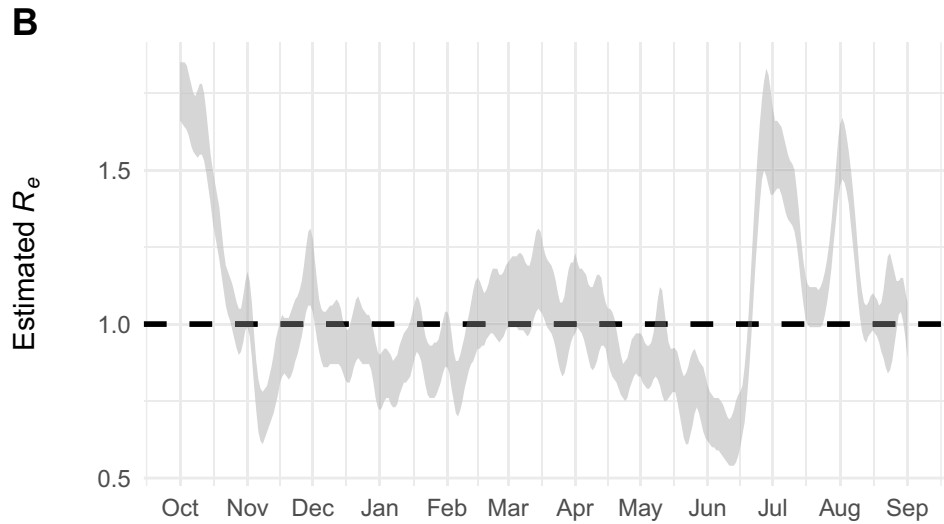

**Fig 3. SARS-CoV-2 epidemic in Switzerland from October 2020 to September 2021.** A: Number of laboratory-confirmed SARS-CoV-2 cases per day. B: Effective reproduction ($R_e$) number of SARS-CoV-2 in Switzerland.

Immunitas study and set to 15% and 25% of the overall population for 1 October 2020 and 1 February 2021, respectively [37]. The initial numbers of exposed and infectious individuals infected with previously circulating variants were based on the number of laboratory-confirmed SARS-CoV-2 cases, the ascertainment rate, and the infectious period. The initial number of exposed and infectious individuals infected with the respective VoCs was set to zero. All other state variables, unless otherwise specified, were set to zero. Model simulations were performed in R (version 4.0.3) and code files are available on the following GitHub repository: github.com/ISPMBern/voc_imports_ch.

**Counterfactual scenarios.** We evaluated the potential impact of different intervention strategies to mitigate the spread of VoCs in Switzerland in various counterfactual scenarios:

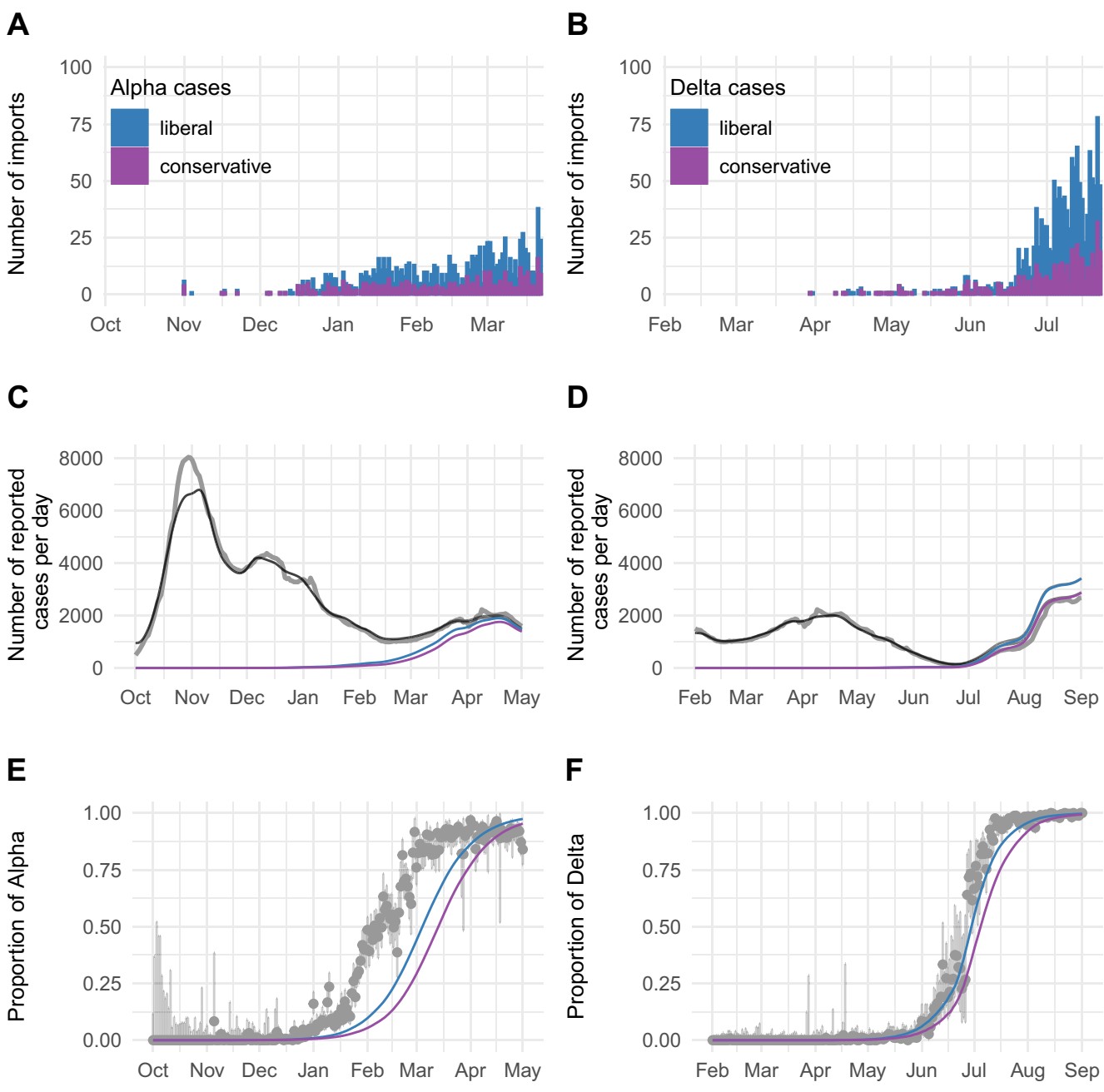

**Fig 4. Dynamics of Alpha and Delta importation to Switzerland.** A, B: Number of imports estimated with the phylogenetic analysis. C, D: Number of laboratory-confirmed SARS-CoV-2 cases per day (gray). The black lines show the overall simulated number of reported cases. The blue and purple line show the simulated number of VoC cases. E, F: Proportion of reported Alpha and Delta among all SARS-CoV-2 infections. Gray: Genomic surveillance data. Blue: Liberal approach. Purple: Conservative approach.

a. Existing border closure: We assumed that borders were already closed at the time of the first estimated import. We then simulated the opening of borders 1 to 100 days after the first estimated import and introduced imports as estimated from the phylogenetic analysis from that date forward.

**Table 1. Parameter values of the SARS-CoV-2 transmission model.**

| Model parameter | Description | Value | Source |
|---|---|---|---|
| $N$ | Population size | 8,644,780 | Swiss FOPH [27] |
| $R_e$ | Effective reproduction number | Fig 3B | Huisman et al. [28] |
| $\beta_t$ | Transmission rate of previously circulating variant | $\frac{R_e S}{(1+p\kappa)\gamma}$ | Based on $R_e$, where $p = E_2/(E_1+E_2)$ corresponds to the proportion of the imported VoC |
| $\kappa$ | Increased transmissibility of VoC | Alpha: 41% Delta: 56% | Estimated from genomic data [31] |
| $1/\sigma$ | Incubation period | 2.6 days | Based on a generation time of 5.2 days [30] |
| $1/\gamma$ | Infectious period | 2.6 days | Based on a generation time of 5.2 days [30] |
| $1/\zeta$ | Testing delay | 2 days | Assumption |
| $\omega_t$ | Rate of importation of VoCs | Fig 4A and 4B | Based on the phylogenetic analysis |
| $\varepsilon$ | Ascertainment rate | 50% | Informed by Stringhini et al. [38,39] |

Abbreviations: FOPH, Federal Office of Public Health; VoC, variant of concern.

b. Implemented border closure: We assumed that borders were closed on the day of the international warning about the VoCs. For Alpha and Delta, we stopped imports for 1 to 60 days after 16 December 2020 and [40] 24 May 2021 [41], respectively.

c. Increased surveillance of travelers: We assumed improved surveillance, quarantine, and isolation of travelers. We performed simulations where we reduced the number of simulated imports by randomly selecting 1 to 99% of the estimated imports from the phylogenetic analysis.

We calculated the time to dominance (>50%) of the VoC and compared it to the time to dominance as observed in Switzerland. Counterfactual scenarios that did not reach dominance within the predefined period were excluded from the analysis.

**Stochastic model.** We used a generic branching process model to investigate the impact of stochastic effects during the early growth phase of variants. The model accounts for superspreading of SARS-CoV-2 and simulates epidemic trajectories with 1, 10, and 100 seeds, which can be interpreted as imports. The branching process was based on a negative binomial distribution to describe the number of secondary cases, with a mean corresponding to $R_e$ and overdispersion parameter $k$ [13,42,43]. The generation time was sampled from the gamma distribution with a mean of 5.2 days and a standard deviation of 1.72 days [30]. For $R_e$ of the variant, we randomly drew $10^4$ values from a uniform distribution between 1.05 to 1.15. For each seeding scenario, we simulated $10^4$ epidemic trajectories.

## Results

In autumn 2020, the effective reproduction number $R_e$ of SARS-CoV-2 increased substantially which led to a rapid exponential increase in laboratory-confirmed cases in Switzerland (Fig 3A and 3B). In the following weeks and months, cantonal and federal authorities strengthened control measures that led to a reduction of $R_e$ (Fig F(A) in S1 File). On 16 December 2020, researchers in the UK announced a newly discovered SARS-CoV-2 variant with a potentially increased transmissibility (Alpha). In response to these findings, the Swiss federal authorities introduced travel restrictions to travelers from the UK [4] and increased sequencing coverage of SARS-CoV-2 (Figs F(B) and F(C) in S1 File). Nevertheless, Alpha then replaced the previously circulating variants during a period of high incidence from January to March 2021,

pushing $R_e$ above 1 again. Based on genomic sequencing, we estimated that Alpha reached dominance (>50%) in Switzerland on 5 February 2021. In May 2021, Delta was identified as a new VoC. Subsequent growth of Delta in June and July 2021 led to an increase of laboratory-confirmed SARS-CoV-2 cases during summer 2021. We estimated that Delta reached dominance in Switzerland on 27 June 2021. The simulated timing of dominance lagged estimates from sample data more for Alpha than for Delta, which might have been influenced by missing early imports indicated by larger clusters for Alpha than for Delta at the beginning (Fig G in S1 File).

We estimated the number of Alpha and Delta importations to Switzerland using two phylogenetic approaches. The Alpha tree contained 7,988 Swiss Alpha sequences, 9,901 non-Swiss context sequences, and 162 background sequences. The Delta tree contained 5,210 Swiss Delta sequences, 9,973 non-Swiss context sequences, and 147 background sequences. Using the liberal approach, we found 1,038 and 1,347 imports of Alpha and Delta into Switzerland, respectively (Fig 4A and 4B and Table A in S1 File). With the conservative approach, we found 383 and 455 imports of Alpha and Delta into Switzerland, respectively. In our sensitivity analysis where we removed half of the available non-Swiss sequences before identifying the most genetically similar sequences to our Swiss sequences, the liberal and conservative approach had similar results as using the baseline sampling method (Fig H in S1 File).

We then introduced these estimated imports in the deterministic SARS-CoV-2 transmission model, which resulted in 593,418 and 592,768 simulated reported cases from 1 October to 1 May with the liberal and conservative approach, respectively, compared to 606,575 cases reported by the FOPH (Fig 4C and Table A in S1 File). For 1 February to 1 September, we simulated 288,397 and 271,702 SARS-CoV-2 cases with the liberal and conservative approach, respectively, compared to 260,933 cases reported by the FOPH (Fig 4D and Table A in S1 File).

To better understand the dynamics of VoCs replacing previously circulating variants, we estimated the time VoCs reached dominance. With the liberal approach, Alpha reached 50% of all infections on 5 March 2021 and Delta reached 50% on 30 June 2021 (Fig 4E and 4F). With the conservative approach, dominance was reached somewhat later, namely on 22 March 2021 and 9 July 2021, respectively. Compared to genomic monitoring data, the model lagged 28 to 45 days behind for Alpha and 3 to 12 days behind for Delta, respectively. This suggests that either the liberal approach better approximates the true number of imported cases that resulted in subsequent transmission chains, that certain events during the early phase of importation accelerated the growth of VoCs, or both. Stochastic simulations of imported variants highlight that the expected variation in the early growth phase can shift the epidemic trajectories by several weeks (Figs I(A) and I(B) and I(C) in S1 File).

We investigated different counterfactual scenarios to assess the impact of border closures and increased surveillance of travelers on the early spread of VoCs. Existing border closures at the time of the first estimated import could delay the time to dominance from a few days to several weeks (Fig 5A). Longer border closures result in increasing returns, e.g., doubling the time of closed borders from 50 to 100 days increases the time to dominance roughly 4-fold. Complete border closure for 100 days delayed the time to dominance by around 40 days. The effect of closed borders is substantially reduced when implemented after the international warning (Fig 5B). Increased surveillance of travelers that reduces the number of imported VoCs that result in a subsequent transmission chain by 25%, 50%, and 75% would delay the time to dominance of Alpha for 4, 10, and 19 days (95% compatibility interval (CI): 2–6, 7–14, and 13–28 days) and of Delta for 3, 6, and 14 days (95% CI: 2–5, 5–10, and 8–18 days) (Fig 5C).

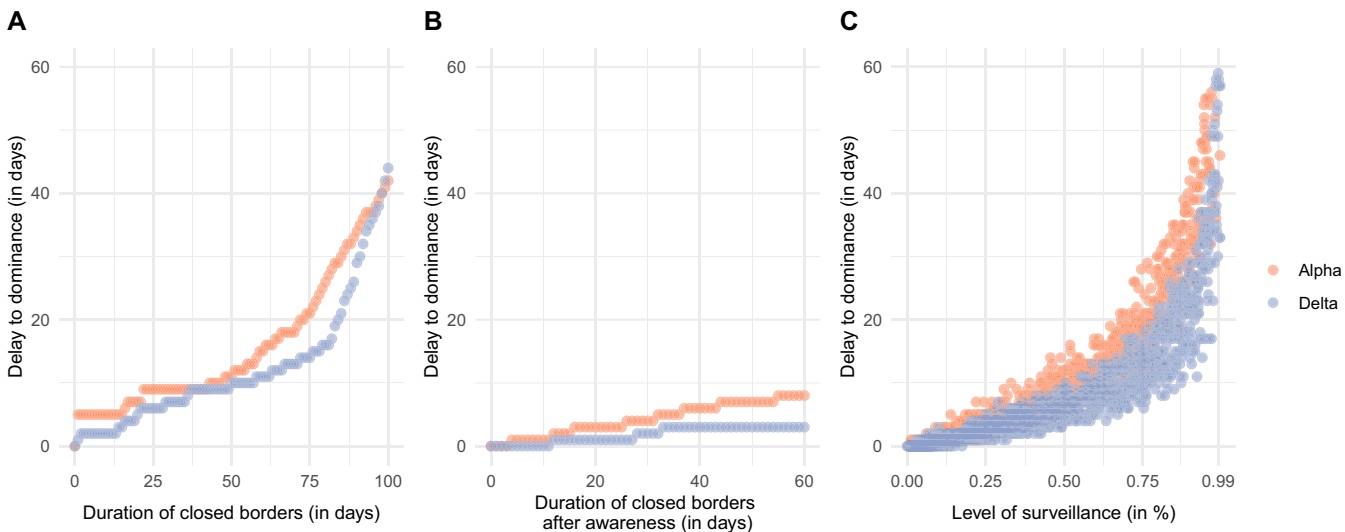

**Fig 5. Counterfactual scenarios for mitigating the spread of VoCs in Switzerland.** A: Existing border closure. B: Implemented border closure after warning about VoC. C: Increased surveillance of travelers.

## Discussion

We used a combination of phylogenetic analysis and dynamic transmission modeling to estimate the number of imported VoCs and simulate the impact of counterfactual intervention scenarios in Switzerland. In the phylogenetic analysis, we found that single importation events happened early and at least several hundred Alpha (383–1,038) and Delta (455–1,347) cases were introduced into the Swiss SARS-CoV-2 epidemic during the study period. The actual number of imports is likely between these estimates. From our sensitivity analysis, we saw that even if there had been substantially less sequencing outside of Switzerland to help infer imports, we would still detect around the same number of introductions. The integration of these importation events into a transmission model accurately described the subsequent spread of VoCs. We estimated a 41% and 56% increased transmissibility of Alpha and Delta compared to previously circulating variants, which is in the range of previously reported estimates [5,7–11]. Applying our transmission model using counterfactual intervention scenarios showed that only very strict or existing control measures would substantially delay the time to dominance of VoCs. In contrast, implementing border closures after international warnings delayed the time to dominance of VoCs by a few days only. Increased surveillance of travelers—which is less disruptive than border closures—could prove effective for delaying the spread of VoCs in situations where their severity remains unclear. These findings have important implications for informing intervention strategies in the case of newly emerging SARS-CoV-2 VoCs and future pandemic preparedness.

The major strength of our study is the combination of phylogenetic analyses with a dynamic transmission model. This not only allowed us to estimate the number of imports but also to investigate counterfactual intervention scenarios. Our phylogenetic approach to estimate imported variants is standardizable and thus applicable to other countries with similar genomic surveillance and other emerging pathogens.

Our study also has a number of limitations. First, we used a deterministic transmission model for our main analysis and ignored the potential effects of stochasticity during the importation and early growth phase of VoCs. In the model, we assumed that all imported VoCs enter a deterministic growth trajectory. In contrast, some imported VoCs might not

result in continuous transmission chains and go extinct even with $R_e > 1$. We showed that stochastic effects during importation and the early growth phase can cause a variation in the time to dominance of several weeks. With the phylogenetic analysis, we found sufficiently large transmission clusters and imports of VoCs that were successfully established in the local population. Second, we fixed the generation time for Alpha, Delta, and earlier circulating variants to 5.2 days. There is some evidence that the generation time of Delta is somewhat shorter [44], but we do not expect this to affect our results substantially. Third, we did not consider the importation of other variants that could compete with the new VoCs. Since the new VoCs were characterized by an increased transmissibility, we do not expect this assumption to substantially affect the dynamics of the new VoCs replacing previously circulating variants. Fourth, we did not consider co-infection with different variants [45]. As we focus our analysis on the spread of VoCs and not on their evolution, we do not expect this to affect our results substantially. Fifth, we assumed a constant ascertainment of SARS-CoV-2 infections of 50%, which was informed by seroprevalence studies [38,39]. During the study period, test positivity varied and ascertainment might have fluctuated as well. Thus, our transmission model cannot precisely describe the overall number of infections, which was not the objective of our study. In addition, the high variation in genomic sequencing among different countries can influence the estimated number of imports from the phylogenetic analysis. For example in Switzerland the sequencing coverage increased from 2% in December 2020 (just after the first importation of Alpha) to 10% in April 2021 (during the first importation of Delta) [46,47], and was on average 14% for our study period (1 October 2020 to 1 September 2021). The limited sequencing efforts during the introduction of the Alpha variant may hinder the detection of early imports and splitting clusters accurately. Sixth, the lag time of detecting an import and the true introduction might vary by testing and sequencing strategies, e.g., du Plessis et al. (2021) reported a detection lag of 3 to 13 days of the introduced cluster time of the most recent common ancestor (TMRCA) to the true importation [6]. Finally, the phylogenetic analysis has intrinsic limitations due to the fact that not every infection is detected or sequenced, and interpretation during outbreaks requires caution [2,48], which was also highlighted at the beginning of the pandemic by Morel et al. 2020 [49]. This is specifically true for Alpha, which appeared at the end of 2020, prior to sequencing in Switzerland being scaled up considerably during 2021. To further mitigate the impact of sequencing error and unreliable sequences, as highlighted by Turakhia et al. (2020) [50], we excluded sequences with poor coverage and low quality control scores and masked homoplastic sites. There is currently no gold standard for estimating imports. Uncertainty in creating and interpreting phylogenies can lead both to imports being underestimated due to considering only one importation event per cluster and overestimated due to within-country diversification or further exports. In our analysis, we aimed to address this issue by using two approaches to estimate lower and upper bounds on the number of imports. The liberal approach considered any cluster containing Swiss sequences as an importation event. If the sequencing coverage was low, the liberal approach would thus overestimate imports and exports could cause Swiss sequences to be falsely classified as imports. In contrast, the conservative approach considered only the earliest clusters with Swiss sequence as imports, which means that further potential imports were strictly excluded.

For monitoring the transmission dynamics of SARS-CoV-2, it is important to differentiate between locally-acquired and imported infections, particularly in situations when the local incidence of infections is low [13]. Tomba and Wallinga (2008) emphasized the significance of estimating the impact of travel on infectious disease transmission and the potential effectiveness of interventions on the local spread, particularly highlighting that initial importation events will eventually lead to a local epidemic if not contained, whereas importations after local expansion has begun are less impactful [51]. Using an example of pandemic influenza,

they estimated that a 90% reduction of imports would delay the first importation event by 11.5 days. Based on routine surveillance data from FOPH, we previously extrapolated a total of 6,211 and 37,061 imported cases in Switzerland during summer 2020 and 2021, respectively. Using genomic data to estimate the number of imports might have the advantage of being less prone to the bias in routine surveillance data. Other studies also using phylogenetics estimated the number of imported infections during the early phase of the SARS-CoV-2 pandemic. For example, 13 and 101 introductions were estimated to South Africa over two and eight months [26,52], 120 introductions were estimated in Boston over three months [23], and more than 1,000 introductions were estimated to the UK over three months [6]. During summer 2020, we estimated 34 to 291 introductions of the SARS-CoV-2 variant EU1 to Switzerland over four months [3]. Phylogenetics offers an advantage in estimating imports by leveraging sequence data that was readily produced by many countries in the pandemic, but can be limited by insufficient coverage to detect importations, fluctuations in coverage, and dependence on other countries to also sequence sufficiently. Methods such as that employed by Pung et al. in Singapore can incorporate detailed case finding and contact tracing data to break down the impact of different measures even further, but the availability of such data varies widely across countries and pandemic stage [53].

Global genomic surveillance is essential to monitor the emergence and spread of VoCs. As of February 2023, more than 14 million SARS-CoV-2 sequences have been submitted to GISAID [33]. This effort facilitated the early identification of several VoCs, such as Omicron in November 2021 [52]. Early detection of variants with substantial immune evasion or altered severity can inform policy makers to adjust control measures, vaccination programs, and health systems. For example, several countries imposed controversial travel bans for visitors from South Africa to prevent importation of Omicron [54]. In our analysis, we found that complete border closures following warnings of VoCs have limited impact and delay their spread by a couple of days only. Hence, travel bans and any time they may buy for certain interventions, such as increasing the uptake of booster vaccines, have to be carefully balanced against the societal and economic costs that accompany them. Similar to our findings, McCrone et al. reported that control measures that had just been introduced in response to warnings about Delta were late, as introductions have already occurred to other countries [24]. In addition to studying importation and exportation events of VoCs, it is important to better understand the spread of VoCs in the local context. Variations in naturally-acquired and vaccine elicited levels of immunity, local control measures, mobility, and behavior can strongly influence the local spread of VoCs. Taking these factors into account will be critical to inform country-specific strategies to respond to the emergence of new VoCs.

Integrating phylogenetic analyses with dynamic transmission models can provide critical insights into the importation and early spread of SARS-CoV-2 VoCs, and how they are impacted by different intervention scenarios. In this study, we showed that border closures would have had a limited impact on the spread of Alpha and Delta in Switzerland. In contrast, increased surveillance of travelers can potentially delay the spread of VoCs by several weeks, which can buy time for health systems to prepare for new epidemic waves.

## Supporting information

**S1 File. Method of the phylogenetic approach and supporting figures for the main text.** (DOCX)

**S1 Table. Data availability statement to use data from GISAID.** (DOCX)

**S2 Table. Accession number for SARS-CoV-2 genomes sequenced in Switzerland.**
(PDF)

## Acknowledgments

We gratefully acknowledge all data contributors, i.e., the authors and their originating laboratories responsible for obtaining the specimens, and their submitting laboratories for generating the genetic sequence and metadata and sharing via the GISAID Initiative, on which this research is based. We also gratefully acknowledge and thank all labs around the world that have collected and shared SARS-CoV-2 sequences we used in our study. A complete list of the labs that generated the data we used from GISAID can be found at doi.org/10.55876/gis8.221003xn. In particular, we want to especially thank the Swiss laboratories that perform SARS-CoV-2 sequencing, namely 'Hôpitaux Universitaires Genève' (HUG), the 'Centre hospitalier universitaire vaudois' (CHUV), the 'Universitätsspital Basel', the 'Institut für Infektionskrankheiten' (IFIK) of the University of Bern, 'Institute of Medical Virology' (IMV) of the University of Zurich, the 'Ente Ospedaliero Cantonale' (EOC) in Bellinzona, and 'Institut Central des Hôpitaux du Valais'.

## Author Contributions

**Conceptualization:** Martina L. Reichmuth, Emma B. Hodcroft, Christian L. Althaus.

**Data curation:** Martina L. Reichmuth, Emma B. Hodcroft.

**Formal analysis:** Martina L. Reichmuth, Emma B. Hodcroft.

**Funding acquisition:** Emma B. Hodcroft, Christian L. Althaus.

**Investigation:** Martina L. Reichmuth, Emma B. Hodcroft.

**Methodology:** Martina L. Reichmuth, Emma B. Hodcroft, Christian L. Althaus.

**Project administration:** Martina L. Reichmuth.

**Resources:** Martina L. Reichmuth, Emma B. Hodcroft, Christian L. Althaus.

**Supervision:** Christian L. Althaus.

**Validation:** Martina L. Reichmuth, Emma B. Hodcroft.

**Visualization:** Martina L. Reichmuth.

**Writing – original draft:** Martina L. Reichmuth.

**Writing – review & editing:** Martina L. Reichmuth, Emma B. Hodcroft, Christian L. Althaus.

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
