## [Decision Letter · Decision Letter 0]

3 Apr 2023

Dear MSc Reichmuth,

Thank you very much for submitting your manuscript "Importation of Alpha and Delta variants during the SARS-CoV-2 epidemic in Switzerland: phylogenetic analysis and intervention scenarios" for consideration at PLOS Pathogens. As with all papers reviewed by the journal, your manuscript was reviewed by members of the editorial board and by several independent reviewers. In light of the reviews (below this email), we would like to invite the resubmission of a significantly-revised version that takes into account the reviewers' comments.

We cannot make any decision about publication until we have seen the revised manuscript and your response to the reviewers' comments. Your revised manuscript is also likely to be sent to reviewers for further evaluation.

Sincerely,

Shuo Su

Academic Editor

PLOS Pathogens

Ronald Swanstrom

Section Editor

PLOS Pathogens

Kasturi Haldar

Editor-in-Chief

PLOS Pathogens

orcid.org/0000-0001-5065-158X

Michael Malim

Editor-in-Chief

PLOS Pathogens

orcid.org/0000-0002-7699-2064

Reviewer's Responses to Questions

**Part I - Summary**

Reviewer #1: This paper attempts to combine compartmental models of epidemiology and transmission graphs derived from phylogenetic

analysis to test various policy counterfactuals and their effects. In general, the paper is vague on details regarding

the phylogenetic analysis, and therefore fails to be a convincing argument. My specific issues are detailed below, but

in short the authors took an inferred phylogenetic tree as ground truth to perform their analyses. While this is a

normal assumption in many other papers, it is particularly problematic when applied to SARS-CoV-2 data. Therefore, I

request that the authors improve the robustness of their analysis by considering phylogenetic uncertainty (some

suggestions have been listed below).

I cannot speak to the portion of analysis based on compartmental models, however it seems to my eyes to also be quite

vague on details.

At the moment, there is nothing to my eyes that would constitute unfixable errors in methodology. Furthermore, the

testing of policy interventions is self-evidently valuable, regardless of the particular conclusions. However, the

amount of work required to make those conclusions robust and reliable might be significant, as the authors made

assumptions about their data (that phylogenetic analysis is reliable in this case) which are not born out by prior work.

Reviewer #2: Summary

The authors combine phylogenetic inference and mathematical models to explore the impact of border closure to prevent the spread of SARS-CoV-2 variants in Switzerland. The phylogenetic inference was performed on sequences from the Alpha and Delta variant. By inferring ancestral states, the authors could estimate a conservative and a liberal number of introductions. They then used these estimates to parameterise an SEIR transmission model with two viral strains (a resident and a mutant) and performed counterfactual scenarios to explore the impact on the time until variant dominance of closing border for a given duration or increasing surveillance.

I found the phylogenetic inference impressive and the mathematical modelling robust. The combination of the two is promising and, generally, the authors are also very aware of the limitations of their approach. However, I still have a few concerns or questions. The main suggestion is to better demonstrate the added value of combining phylogenetic inference and compartmental modelling, for instance by performing sensitvity analyses but also by better discussing earlier studies on similar topics.

**Part II – Major Issues: Key Experiments Required for Acceptance**

Reviewer #1: The citations in the PDF provided are essentially dead links. If clicked on, the link directs the user to a Zotero page

with no meaningful information. This should be corrected before the final publication, and it makes reading the paper

and checking the references a bit difficult.

The description of the conservative and liberal approaches to count imports is quite vague. For the conservative

estimate, it isn't clear what "counts" as a subtree. The Figure 1A suggests that it is some maximal clade, which

contains all the Swiss sequences. However, if this is the case, then by

> When Swiss and non-Swiss sequences intermix within a subtree, the conservative approach count[s] this as only one

> import.

the conservative estimate would always be 1, for any tree. However, Figure 4A and B show conservative counts much higher

than 1, so there must be some method to picking the subtrees other than "maximal clades" which is not described in the

paper.

The liberal approach seems to be more clear, as it seems to be counting the number of "monophyletic" clades in the tree,

and probably includes individual tips. In this case, it seems it would be the minimum number of clades s.t. all clades

are "monophyletic" and all Swiss sequences are accounted for. Of course, this definition can be made more rigorous by

describing the clade as sets or partitions on tips. Nonetheless, as this metric is an important result of the paper, a

explicit and rigorous definition should be presented.

Secondly, the above definitions are in conflict with the first sentence:

> We inferred an import whenever a node without Swiss sequences led to a node with Swiss sequences.

Which is

a) very vague as there is no description of how internal nodes are assigned to be either Swiss or non-Swiss and

b) contradictory with later definitions.

Basically, the section describing this section needs to be made much more clear.

However, regardless of the particular definition of conservative and liberal estimates of imports, performing this

operation on a particular tree is methodologically questionable. Recall that phylogenetic analysis of SARS-CoV-2 data is

difficult[1], and the phylogenies are unstable [2]. Sequences which are sampled close together often differ by only a

sites, leading to the case where the support for any _particular_ phylogeny is not very strong. Furthermore, sequencing

error from some labs might bias the results in this case [3]. The phylogeny in this paper appears to be constructed via

the Nextstrain pipeline [4], which is well regarded. However, this pipeline doesn't escape the fundamental problems that

there doesn't seem to be enough signal to reliably resolve the phylogeny to the detail required for this paper.

This difficulty in resolving the fine grained phylogenetic relationships between sequences is a particular issue for

this paper, as a core metric relies on accurately and reliably being able to resolve these relationships. I hesitate to

give a specific instruction to fix methodology, but in this case _some_ level of uncertainty analysis must be done. The

authors could construct a tree set like in [1], or to sample trees from the posterior in some Bayesian inference program

(such as BEAST or RevBayes). In either case, the authors should rerun their analyses on these sample trees, in order to

ensure that the results are stable. It is possible that the conclusions of this paper hold, however it is not clear at

this point that the conclusions of this paper are dependant on a particular (and possibly incorrect) tree.

[1]: https://doi.org/10.1093/molbev/msaa314

[2]: https://doi.org/10.1371/journal.pgen.1009175

[3]: https://virological.org/t/issues-with-sars-cov-2-sequencing-data/473

[4]: Though this is never explicitly stated, and can only be inferred via a reference to another paper. However, that

paper has procedures for sub sampling the dataset which do not directly translate to the subject matter at hand. In

particular, the number of sequences reported to be in the tree in Hodcraft 2021 is different than then number of

sequences reported in this paper.

Reviewer #2: 1) Sampling rate

The authors are working with an absolute number of introductions in the country. Intuitively, this number seems very dependent on the sampling rate (proportion of the infections sequenced), although potentially not in a linear way. Furthermore, as the authors accurately acknowledge in the Discussion, the nature of the dataset used as a reference may also affect the results. On both these topics, the sensitivity analyses performed seem a bit light (or absent).

First, the authors could subsample their dataset to see how the number of introductions scales with the subsampling. I think this can be done easily because it does not require to re-infer the phylogeny. Second, the authors could try a few different reference datasets to show that their estimates remain unaffected. For the latter, the computational burden is higher so perhaps a few datasets will suffice.

2) Model results

One of the main results is that the model underestimates the increase in variant frequency, even when using the liberal definition of importation events. This lag is less pronounced for Delta than for Alpha. In my opinion, this can either come from the importation rate or from some of the model specifications. At any rate, the origin of this mismatch and why it is more pronounced for the Alpha variant should be explored more thoroughly. One possibility could be to use the observed data to infer the VOC import rate instead of fixing it (unless another parameter seems less reliable).

3) Model parameterisation

As earlier studies, e.g. Du Plessis et al (Ref. 6 in the manuscript, which could perhaps be discussed some more), the authors can infer the number but also the date of VOC introduction. I wonder if turning this into a constant input rate is not throwing some of the information away. Put differently, if the goal is to infer such a rate while all the other model parameters are known (see Table 1), then why not use a more appropriate dataset and, for instance, try to fit the proportion curves?

3) Insights

Overall, I found that the discussion about the insights from the study was somehow limited. The most interesting bit was the second to last paragraph from line 361 about Omicron. One way to improve the discussion could be to refer to earlier studies. For instance, a quick search online pointed towards a paper from Scalia Tomba & Wallinga (2008, Math Biosci) with an eloquent title: "A simple explanation for the low impact of border control as a countermeasure to the spread of an infectious disease". More recently, Pung et al (2023, BMC Med) seem to have developed a more detailed model addressing similar questions.

Related to this point, showing the added value of fitting the number of introductions from the sequence data rather than on the relative variant proportion might help. For this, discussing differences with other studies that performed similar inferences, e.g. Du Plessis et al in England, seems important.

4) Data sharing

Before criticising, I want to stress that the authors made impressive efforts to share the code they developed! Perhaps they could also consider a more permanent repository than GitHub (either PLoS Pathogens supplementary materials or a repository with a DOI).

However, given the current context about GISAID usage, it seems important to check that the laboratories who contributed essential data to this work (so the Swiss Alpha and Delta sequences) were contacted to be involved in the study or that there is a national agreement about genetic data usage (akin to the UK genomics consortium). In any case, the sentence in line 388 suggesting that the data is openly shared by GISAID is inaccurate: there are restrictions to this use.

**Part III – Minor Issues: Editorial and Data Presentation Modifications**

Reviewer #1: Honestly, this section is quite short. I stopped adding to it after I realized that significant changes to the methodology would have to be made.

Page 4, starting on 116:

> When Swiss and non-Swiss sequences intermix within a subtree, the conservative approach counted this as only one

> import, with further non-Swiss sequences assumed to originate from parallel evolution outside of Switzerland or

> exports from Switzerland (Fig 1A).

The language here is a bit awkward, and I think there are some error in tenses. "Intermix" is present tense, but

"counted" is past tense, and they should be the same tense, probably present tense.

Page 4, starting on 119: Importation -> import

Page 7, line 177: Importation -> import or "simulated the importation" -> "simulated importation"

Reviewer #2: line 48: the term "lockdown" should be handled with care because it has different meanings in different countries or contexts. Perhaps the authors should spend a bit more time defining it carefully to avoid potential misunderstandings.

line 157: I think kappa estimates a growth advantage rather than a transmission advantage (if a variant causes longer infections with a similar transmission rate it would also be captured). Perhaps update the formulation?

Table 1: I did not understand how the testing delay came into the model (to go from "tested" to "recovered").

page 9: It seems important to mention that the Alpha variant was first detected and studied because of the S-target gene failure. Sequencing confirmed the detection (showing that it was a variant) and was used later to study variant spread.

Figure 5: the nature of the x-axis is unclear (is it how long the border is closed?). Also, having all three panels on a line would help comparisons. Finally, I do not understand why panel c could not be a deterministic model as well.

line 301: specify against which strain the transmission advantage is computed.

Reference 37 (about computing variant growth advantage) is missing.

PLOS authors have the option to publish the peer review history of their article (what does this mean?). If published, this will include your full peer review and any attached files.

Reviewer #1: **Yes: **Ben Bettisworth

Reviewer #2: **Yes: **Samuel Alizon
---

## [Decision Letter · Decision Letter 1]

20 Jun 2023

Dear MSc Reichmuth,

Thank you very much for submitting your manuscript "Importation of Alpha and Delta variants during the SARS-CoV-2 epidemic in Switzerland: phylogenetic analysis and intervention scenarios" for consideration at PLOS Pathogens. As with all papers reviewed by the journal, your manuscript was reviewed by members of the editorial board and by several independent reviewers. The reviewers appreciated the attention to an important topic. Based on the reviews, we are likely to accept this manuscript for publication, providing that you modify the manuscript according to the review recommendations.

Sincerely,

Shuo Su

Academic Editor

PLOS Pathogens

Ronald Swanstrom

Section Editor

PLOS Pathogens

Kasturi Haldar

Editor-in-Chief

PLOS Pathogens

orcid.org/0000-0001-5065-158X

Michael Malim

Editor-in-Chief

PLOS Pathogens

orcid.org/0000-0002-7699-2064

Reviewer Comments (if any, and for reference):

Reviewer's Responses to Questions

**Part I - Summary**

Reviewer #1: The authors have done an excellent job addressing the (possibly overly callous on my part) comments from the reviewers.

At this time, I only have minor issues with the added explanation regarding the explanation of how imports are

calculated.

Reviewer #2: I thank the authors for carefully answering my concerns. I found the new sensitivity analysis on the importance of sampling particularly thorough. The model specifications are also clearer and I realised that I misunderstood the definition of the import rate. Overall, I think the authors now better show what the phylogenetic component brings to these counterfactual models.

**Part II – Major Issues: Key Experiments Required for Acceptance**

Reviewer #1: All of my major concerns have been addressed. I quite like the addition of the sensitivity analysis, and I think it

greatly improves the reliability of the results presented in the paper.

Reviewer #2: My sole suggestion is still about this sampling rate because, if I think I now understand how it plays in the model, it might still not be the case. More precisely, in the Methods, \\omega_t is never really properly defined. All that is said is that it "was based on the daily number of estimated imports" (page 5). Introducing it from the section "Phylogenetic analysis" (pages 3-4) would help insist on its added value. Furthermore, this time-varying property could also be underlined in the results because currently the authors only mention the total number of introductions ("we found 1,038 and 1,347 imports of Alpha and Delta into Switzerland, respectively", page 8). How did this input rate vary with time? Was it proportional to the circulation of the VOC in Europe or in the world? Addressing these questions by leaning on the (beautiful) Figure 4 would further show the importance of phylogenetic insights.

**Part III – Minor Issues: Editorial and Data Presentation Modifications**

Reviewer #1: The added explanation for how the number of imports is an improvement, but I still feel it is a bit unclear, especially

after reading the supplement. However, I think that this can be fixed easily. On line 37:

"we collapsed subtrees that contain only sequences from a single country into the parental node recursively" ->

"we collapsed subtrees which contain only sequences from a single country into the parent node to form a polytomy. This

process was repeated in a recursive 'bottom-up' fashion, such that every node eligible for collapse was collapsed."

And add "we classified inner nodes based on the composition of their _direct_ children, after collapsing subtrees into

polytomies" (or something like that, I don't mean to dictate your voice) to the same paragraph.

Reviewer #2: page 7: The SGTF sentence might be better suited in the introduction but it's up to you.

page 10: Do you mean that Tomba and Wallinga (2008) showed that you need a 90% or greater reduction in imports so that the time to dominance can be delayed by more than a week?

PLOS authors have the option to publish the peer review history of their article (what does this mean?). If published, this will include your full peer review and any attached files.

Reviewer #1: No

Reviewer #2: **Yes: **Samuel Alizon

Figure Files:

Data Requirements:

Reproducibility:

References:

---

## [Decision Letter · Decision Letter 2]

11 Jul 2023

Dear MSc Reichmuth,

We are pleased to inform you that your manuscript 'Importation of Alpha and Delta variants during the SARS-CoV-2 epidemic in Switzerland: phylogenetic analysis and intervention scenarios' has been provisionally accepted for publication in PLOS Pathogens.

Best regards,

Shuo Su

Academic Editor

PLOS Pathogens

Ronald Swanstrom

Section Editor

PLOS Pathogens

Kasturi Haldar

Editor-in-Chief

PLOS Pathogens

orcid.org/0000-0001-5065-158X

Michael Malim

Editor-in-Chief

PLOS Pathogens

orcid.org/0000-0002-7699-2064

Reviewer Comments (if any, and for reference):

Reviewer's Responses to Questions

**Part I - Summary**

Reviewer #1: I have no further issue with the manuscript, and I think it is ready for publication.

Reviewer #2: (No Response)

**Part II – Major Issues: Key Experiments Required for Acceptance**

Reviewer #1: (No Response)

Reviewer #2: (No Response)

**Part III – Minor Issues: Editorial and Data Presentation Modifications**

Reviewer #1: (No Response)

Reviewer #2: (No Response)

PLOS authors have the option to publish the peer review history of their article (what does this mean?). If published, this will include your full peer review and any attached files.

Reviewer #1: No

Reviewer #2: **Yes: **Samuel Alizon

---

## [Editor Report · Acceptance letter]

6 Aug 2023

Dear MSc Reichmuth,

We are delighted to inform you that your manuscript, "Importation of Alpha and Delta variants during the SARS-CoV-2 epidemic in Switzerland: phylogenetic analysis and intervention scenarios," has been formally accepted for publication in PLOS Pathogens.

Best regards,

Kasturi Haldar

Editor-in-Chief

PLOS Pathogens

orcid.org/0000-0001-5065-158X

Michael Malim

Editor-in-Chief

PLOS Pathogens

orcid.org/0000-0002-7699-2064